

# Comment on "An approach to sulfate geoengineering with surface emissions of carbonyl sulfide" by Quaglia et al. (2022)

Marc von Hobe[1], Christoph Brühl[2], Sinikka T. Lennartz[3], Mary E. Whelan[4], Aleya Kaushik[5,6]

[1]Institute for Energy and Climate Research (IEK-7), Forschungszentrum Jülich GmbH, 52425 Jülich, Germany
[2]Max Planck Institute for Chemistry, Mainz, Germany
[3]Institute for Chemistry and Biology of the Marine Environment, University of Oldenburg, Carl-von-Ossietzky-Straße 9–11, 26129 Oldenburg, Germany
[4]Department of Environmental Sciences, Rutgers University, New Brunswick, NJ, USA
[5]Cooperative Institute for Research in Environmental Sciences (CIRES), University of Colorado, Boulder, CO, USA
[6]NOAA Global Monitoring Laboratory (GML), Boulder, CO, USA

*Correspondence to*: Marc von Hobe (m.von.hobe@fz-juelich.de)

**Abstract.**   Solar radiation management through artificially increasing the amount of stratospheric sulfate aerosol is being considered as a possible climate engineering method. To overcome the challenge of transporting the necessary amount of sulfur to the stratosphere, Quaglia and co-workers suggest deliberate emissions of carbonyl sulfide (OCS), a long-lived pre-
cursor of atmospheric sulfate. In their paper, published in *Atmospheric Chemistry and Physics* in 2022, they outline two scenarios with OCS emissions either at the Earth's surface or in the tropical upper troposphere and calculate the expected radiative forcing using a climate model. In our opinion, the study (i) neglects a significantly higher surface uptake that will inevitably be induced by the elevated atmospheric OCS concentrations and (ii) overestimates the net cooling effect of this OCS geoengineering approach due to some questionable parameterizations and assumptions in the radiative forcing calculations. In this
commentary, we use state of the art models to show that at the mean atmospheric OCS mixing ratios of the two emissions scenarios, the terrestrial biosphere and the oceans are expected to take up more OCS than is being released to reach these levels. Using chemistry climate models with a long-standing record for estimating the climate forcing of OCS and stratospheric aerosols, we also show that the net radiative forcing of the emission scenarios suggested by Quaglia and co-workers is smaller than suggested and insufficient to offset any significant portion of anthropogenically induced climate change. Our conclusion
is that a geoengineering approach using OCS will not work under any circumstances and should not be considered further.

## 1 Introduction

The idea of climate engineering by stratospheric sulfur injections has received widespread attention since Nobel Laureate Paul Crutzen brought up the idea in an essay in the journal *Climatic Change* (Crutzen, 2006). Besides cautionary voices pointing out potential risks of such an intervention into the climate system (e.g. Irvine et al., 2016; Parker and Irvine, 2018; Pitari et al.,
2014; Robock et al., 2009), a significant hitch is posed by the technological and economic challenge of transporting the nec-essary amount of material to the altitude regions where the scattering particles would actually have an effect (e.g. Lawrence et





al., 2018; Robock et al., 2009; Lockley et al., 2020). To this end, it is only natural to think of carbonyl sulphide (OCS or COS), the most important natural non-volcanic precursor of sulfate aerosol in the stratosphere (Kremser et al., 2016, and references therein). In fact, already a decade before the publication of Crutzen's famous essay, the idea to use artificial OCS emissions

as a means for counteracting global warming was considered by Taubman and Kasting (1995), who came to the conclusion that the associated environmental risks were not acceptable.

Recently, Quaglia et al. (2022) revisited OCS emissions as a potentially cheap and easy-to-implement route to enhance the amount of stratospheric sulfate particles. They employed the University of L'Aquila Climate Chemistry Model (ULAQ-CCM) to investigate the radiative forcing as well as indirect effects on ozone, methane and stratospheric water vapour for two OCS

release scenarios with 40 Tg S a$^{-1}$ emitted at the Earth surface or 6 Tg S a$^{-1}$ emitted in the tropical upper troposphere respectively. The study comes to the overall conclusion that such an increase in OCS emissions may be feasible and produce a favourable radiative forcing, and that it may be considered as a possible alternative to the other sulfur injection and geoengineering methods.

In this comment on the Quaglia et al. (2022) study, we quantitatively address the issue of OCS sinks responding to the higher

atmospheric mixing ratios induced by the two scenarios. For the terrestrial biosphere, the issue was already raised in a comment (Whelan, 2021) during peer review of Quaglia et al. (2022), leading to a disclaimer being added that "*our assumption that the rate of COS uptake by soils and plants does not vary with increasing COS concentrations will need to be investigated in future work*". In Section 2 of this comment, the Simple Biosphere Model version 4 (SiB4, Kooijmans et al., 2021) is used to make quantitative estimates of the additional sink for both scenarios, and potential ecosystem exposure effect to the elevated OCS

concentrations are discussed. In Section 3, we calculate an additional OCS sink by the ocean, which we expect to become undersaturated with respect to the atmosphere with mixing ratios in the ppb range. Besides quantifying additional sink terms, we also question the suggested radiative balance between the indirect aerosol cooling and the direct greenhouse gas warming resulting from the additional OCS. In our understanding, the OCS greenhouse gas forcing should scale approximately linearly with concentration. In Section 4, we explain this reasoning and calculate the scaled radiative forcing from Brühl et al. (2012)

to the higher mixing ratios.

## 2 Enhanced OCS uptake by the terrestrial biosphere

The largest sink of atmospheric OCS is uptake through plant stomata (Whelan et al., 2018). Observations of OCS have been used to estimate stomatal conductance and related variables over terrestrial ecosystems for over a decade (e.g. Campbell et al., 2008). Ambient OCS concentrations have risen slowly during the Industrial Era due to anthropogenic sources; however, the

average ambient concentration has not exceeded 500 parts-per-trillion (ppt) in any long-term record (Campbell et al., 2017), and recent measurements suggest that OCS is no longer increasing (Serio et al., 2022). Localized increases in OCS concentrations that deviate from background concentrations cause some leaf stomata to be more open than they would be otherwise.



Given what we know about terrestrial-OCS exchange, our calculations here suggest that not only would it be Sisyphean to maintain a high ambient OCS but elevated OCS would increase plant stress and likely plant mortality worldwide.

## 2.1 The Simple Biosphere Model

Stomata are small openings on plant leaves that can be more open or closed to facilitate gas exchange between plant leaves and the atmosphere. As carbon in the form of $CO_2$ diffuses into plant leaves, water evaporates out. Plants regulate stomatal opening to balance carbon and water needs. The Simple Biosphere model (SiB4, Haynes et al. (2019); Sellers et al. (1986)) is a fully prognostic mechanistic and process-based land surface model that calculates the stomatal conductance of water ($g_{s,w}$). The SiB4 model is forced with MERRA2 meteorological inputs for global calculations and is well described by Haynes et al. (2020). The calculation of leaf water flux is purposefully simplified to engender a straightforward comparison between baseline and elevated OCS scenarios without anticipated feedback effects, using the theoretical relationships in Seibt et al. (2010):

$$F_w = g_w(w_i - w_a) \qquad (1)$$

where $F_w$ is the flux of water out of the leaf in mol $H_2O$ m$^{-2}$ s$^{-1}$, $g_w$ is the conductance of the leaf to water in mol $H_2O$ m$^{-2}$ s$^{-1}$, $w_i$ is the internal concentration of water (assumed to be saturated) and $w_a$ is the ambient concentration of water vapor outside the leaf in mol $H_2O$ mol$^{-1}$. The conductance to water is the combination of the stomatal conductance of the leaf to water, $g_{s,w}$ and the boundary layer conductance of the leaf to water, $g_{b,w}$, both in mol $H_2O$ mol$^{-1}$.

$$g_w = \left(\frac{1}{g_{s,w}} + \frac{1}{g_{b,w}}\right)^{-1} \qquad (2)$$

The ratio of water conductance to OCS conductance, $R_{w\text{-OCS}}$, was determined theoretically by Seibt et al. (2010) to be 2.01. We can calculate the conductance of OCS, $g_{s,OCS}$, from water conductance via

$$g_{s,OCS} = \frac{g_{s,w}}{R_{w-OCS}} \qquad (3)$$

We expect the effect of elevated OCS to vary from species to species. To calculate the order of magnitude of this effect, we take the conservative estimate of 20% of all species responding strongly to high ambient OCS. We then calculate the increase in water flux induced by increasing the stomatal OCS conductance to 50% in 20% of plant functional types (PFTs). Because of the nature of $R_{w\text{-OCS}}$, this results in a corresponding 50% increase of $g_{s,w}$. It is assumed that the heat flux and associated parameters will not change with increased $g_{s,w}$, although this would obviously not be the case.

## 2.2 Stomatal Response to elevated OCS

The two scenarios in the Quaglia et al. (2022) study invoke surface ambient OCS concentrations of 4.8 ppb and 35.5 ppb, roughly 9.6 and 71 times the current ambient concentration. There are no published experiments where plants have been observed under these high levels of OCS. When Stimler et al. (2010) subjected plants to up to 3 ppb OCS, it appeared that while some plants experienced no stomatal response, others exhibited a 2- to 5-fold increase in stomatal conductance. There was no clear pattern as to which plants responded strongly to elevated OCS.



OCS is irreversibly hydrolysed to $H_2S$ within plant leaves by the enzyme carbonic anhydrase. Endogenous $H_2S$ produced by the plant plays an important role in plant growth and development (Zhang et al., 2017). It is likely that the exogenous $H_2S$

from OCS hydrolysis interacts with these same pathways: Stimler et al. (2010) found that mutant plants without carbonic anhydrase no longer exhibited the stomatal response to increased ambient OCS of their wild type counterparts. The fact remains that signalling stomata to open more widely than evolutionarily-advised plant regulation allows would reduce the water use efficiency of a plant and make drought stress intolerable. Below we make a conservative estimate of this effect under elevated ambient OCS.

Since we have little understanding of which plants would have a large response, we investigate a scenario where 20% of terrestrial plants exhibit a 50% increase in stomatal conductance under high OCS (Figure 1). This may be a very conservative estimate since 40% of plants in the Stimler et al. (2010) study exhibited a strong response to increased ambient OCS. We calculate the original and increased evapotranspiration (water) flux as described in Section 2.1 and find a global mean of 8.8 ±0.99% increased evapotranspiration for the elevated OCS scenario. In areas with typical or increased water stress under

climate change, the added water loss through plants would further limit plant function.

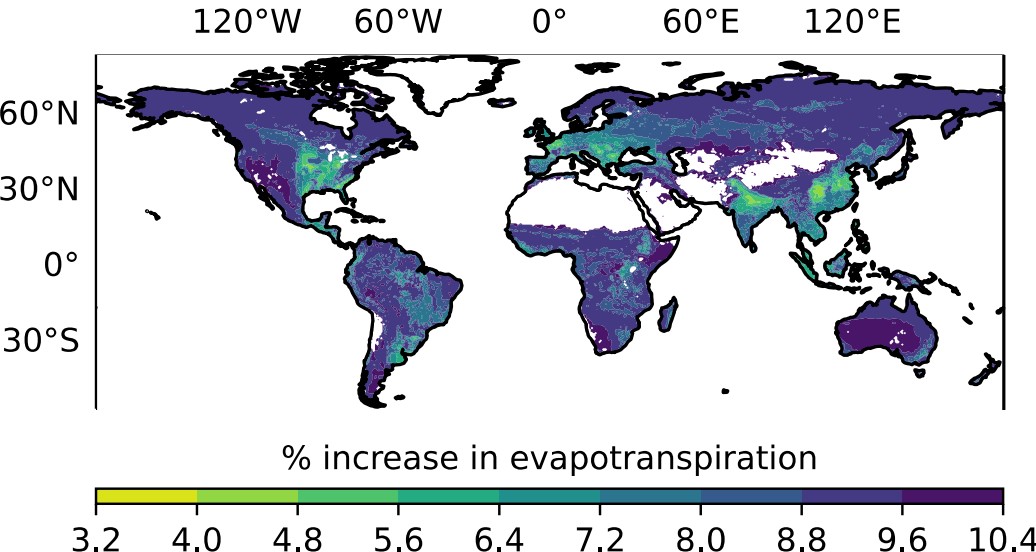

**Figure 1** The average increase in evapotranspiration anticipated under an elevated OCS scenario for the years 2000-2021, where 20% of plants are assumed to experience a 50% increase in stomatal conductance using output from the Simple Biosphere model version 4.2. Areas in white correspond to absence of plant cover, hence SiB4 does not calculate stomatal conductance in those pixels.



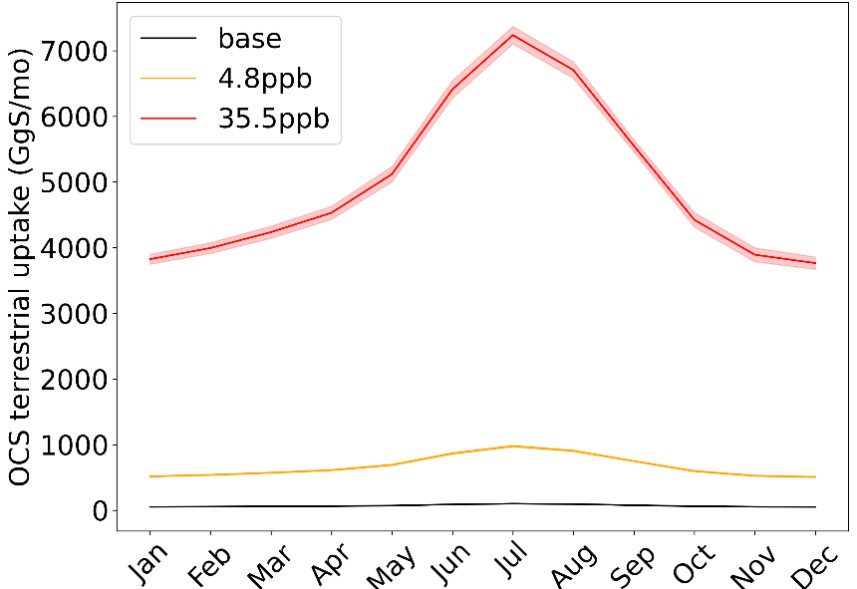

**Figure 2** Simple Biosphere model version 4.2 (SiB4) simulated OCS uptake by plants and soils, per month, at baseline and elevated OCS levels averaged over the years 2000-2021 with 500 ppt OCS (black) and the two OCS geoengineering scenarios with 4.8 ppb (orange) and 35.5 ppb (red).

## 2.3 Uptake by terrestrial ecosystems

A potentially complicated feedback mechanism will determine how much introduced OCS will remain in the atmosphere in actuality: the terrestrial biosphere will take up much of the OCS and the increase in OCS will alter the functioning of the terrestrial biosphere. Feedback mechanisms aside, SiB4 simulates a baseline (0.5 ppb OCS) annual uptake of 0.84 Tg S $a^{-1}$ by plants and soils. When OCS is elevated to 4.8 ppb OCS, the uptake flux is 8.1 Tg S $a^{-1}$; for 35.5 ppb OCS, 59.7 Tg S $a^{-1}$ is taken up by terrestrial ecosystems (Figure 2). This exceeds the corresponding 6 and 40 Tg S $a^{-1}$ released in the proposed geoengineering schemes.

## 3 Enhanced OCS uptake by oceans

In seawater, OCS is produced photochemically (Ferek and Andreae, 1984) and, at slower rates, also in the dark (Von Hobe et al., 2001) from chromophoric dissolved organic matter (CDOM) and destroyed by hydrolysis (Elliott et al., 1987). In most regions, production is slightly dominating, which leads to moderate supersaturations (typically between 1 and 15) and consequently emissions to the atmosphere on the order of 100 Gg S $a^{-1}$ globally (Lennartz et al., 2021). The direction and magnitude



of the OCS flux across the air-sea interface is governed by its concentration gradient between air and water. An increasing atmospheric mixing ratio will shift the solubility equilibrium, and we expect the net flux to reverse at higher atmospheric OCS

levels.

**Figure 3** Regional distribution of annual sea-to-air OCS fluxes for the present day atmosphere with 500 ppt OCS (a) and the two OCS geoengineering scenarios with 4.8 ppb (b) and 35.5 ppb (c).



### 3.1 Uptake calculations

To estimate OCS fluxes between the atmosphere and the ocean for the two OCS emission scenarios considered by Quaglia et al. (2022), we use the same model that was employed by Lennartz et al. (2021) to calculate marine OCS fluxes for the present day atmosphere. This model dynamically calculates OCS concentrations in seawater and prescribes a homogenous atmospheric mixing ratio to calculate emissions (Lennartz et al., 2017). Here we consider three scenarios: the standard scenario as in Lennartz et al (2021) with an atmospheric mixing ratio of 500 ppt, and two scenarios from the Quaglia et al. (2022) paper with 4.8 ppb and 35.5 ppb. Emissions are calculated by

$$F = k \cdot \Delta c = k \cdot \left( c_{water} - c_{equilibrium} \right) = k \cdot \left( c_{water} - \frac{H}{c_{air}} \right) \tag{4}$$

with $k$ being the transfer velocity parameterized by wind speed according to Nightingale et al. (2000).

Maps of the resulting OCS fluxes are shown in Figure 3. The model is driven by satellite products of CDOM (Aqua MODIS, absorption due to gelbstoff and detritus) and meteorological reanalysis products (Lennartz et al., 2021). The model is spun up for 2 years, and results from 2003-2019 are considered for analysis to allow for the effect of interannual variation in ocean productivity. When integrated globally, marine emissions add up to $112.4 \pm 18$ Gg S a$^{-1}$ for the present day atmosphere with 500 ppt OCS and to $1833.0 \pm 20$ Gg S a$^{-1}$ and $15.7 \pm 0.1$ Tg S a$^{-1}$ for the scenarios suggested in the Quaglia et al. (2022) study with 4.8 and 35.5 ppb OCS respectively. In both cases, the ocean becomes a sink for OCS, removing 31 % and 39 % of the respective 6 and 40 Tg a$^{-1}$ OCS added to the atmosphere.

### 3.2 Possible implications in the marine ecosystem

Possible implications of increased atmospheric deposition of OCS for the marine ecosystem are currently unknown. Reversing the direction of the air-sea exchange by increasing the atmospheric concentration of OCS makes hydrolysis the only sink for OCS in the water. Hence, concentrations in seawater are expected to increase as well. Modelled concentrations of OCS in seawater in the scenarios of 4.8 and 35.5 ppb are 40.8 (4.2-450) pmol L$^{-1}$ and 201.3 (10.0-13180) pmol L$^{-1}$ compared to the measured average concentrations of 13.2 (2.3-350) pmol L$^{-1}$ today. Toxicity for marine organisms has not yet been tested explicitly, but ecotoxicological tests on vertebrates have found harmful neurological effects after long-term exposure (Morgan et al., 2004). Proven toxicity for insects and fungi has led to the application of OCS as a pesticide (Zettler et al., 1997). These ecotoxicological tests were conducted at concentrations higher than the ones calculated here, but the effect of long-term exposure of enhanced OCS concentrations should be assessed especially in cold waters with low hydrolysis rates and hence the potential for accumulation of OCS in surface waters, to allow conclusions on implications on the marine ecosystem. Besides the toxicity, increased atmospheric deposition of OCS may alter the sulphur cycle in the surface mixed layer of the ocean. OCS is hydrolysed to $CO_2$ and $H_2S$ (Elliott et al., 1989), the latter oxidizing quickly to sulfate in the oxic mixed layer. However, the overall effect on the sulphur cycle in the mixed layer is expected to be of minor importance in relation to other processes affecting the degradation products of OCS hydrolysis.



## 4 Radiative forcing calculations

When the OCS mixing ratios of Quaglia et al. (2022) for the surface emission scenario were entered into the radiative convective model used in Brühl et al. (2012) the result is an infrared radiative forcing of 0.52 W m$^{-2}$, i.e. compared to the background

OCS increased warming scales about linearly. This value is about 3 times the value estimated by Quaglia et al. (2022). For the profile of the TTL scenario the calculated forcing is 0.15 W m$^{-2}$ (at least).

Concerning shortwave forcing, the value of Quaglia et al. (2022) is overestimated because evaporation of sulfate aerosol in the warm middle and upper stratosphere appears to be neglected in their model. Thermodynamics does not allow for a maximum sulfate concentration at 35 km altitude (their Figure S2). The Junge layer, also the one enhanced by OCS injections, is

below about 31km. In a sensitivity study with the EMAC (ECHAM5/MESSy Atmospheric Chemistry) CCM (e.g. Brühl et al., 2018; Schallock et al., 2023) where 6 Tg S a$^{-1}$ were injected for several years over 5 tropical cities at the tropopause (97 hPa), the resulting additional stratospheric aerosol forcing was found to be about -0.55 +/-0.03 W m$^{-2}$ (top of the atmosphere, solar + thermal). It required about 4 years of continuous injection to reach that approximate plateau value, which depends on the boundary condition at the surface (Fig. 4). The larger greenhouse effect of OCS and smaller thickness of the sulfate layer

considerably decrease the net negative forcing possible by the suggested method for geoengineering. In our calculations we get only -0.4 W m$^{-2}$ for the TTL scenario and present-day conditions (2017 to 2021) which cannot compensate for forcing by anthropogenic $CO_2$.

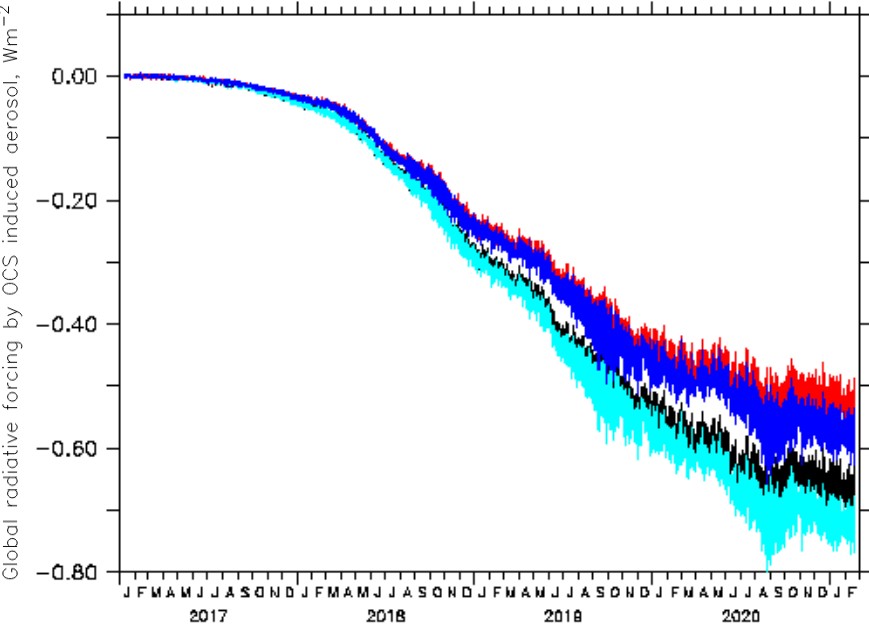

**Figure 4** Globally averaged instantaneous radiative forcing at the top of the atmosphere by aerosol due to continuous OCS injection near the tropical tropopause starting in January 2017 as simulated by EMAC (black and light blue: solar forcing; red and blue: total forcing). For the curves with the smaller absolute values (black and red) the surface mixing ratios of OCS were fixed to observations, for the others (light blue and blue) surface OCS could increase to about 3 ppb from downward transport.



## 5 Conclusions

We have shown that both the terrestrial biosphere and the oceans will respond to higher atmospheric OCS levels under OCS
geoengineering conditions by enhanced uptake. The total additional sink of about 10 and 75 Tg S a$^{-1}$ for both scenarios is
larger than the respective amounts of OCS released, making it questionable that the desired atmospheric concentrations and
radiative effects can even be reached. Even increasing the annual OCS additions of 6 and 40 Tg S a$^{-1}$ by the respective amounts
of additional uptake may not be sufficient because real uptake may be even higher preventing the system from reaching the
target high OCS equilibrium.
Several studies have provided evidence for negative effects of higher OCS exposure and/or uptake on individual plants, insects,
funghi and marine organisms. Such effects need to be understood in more detail and be explored on a wider range of organisms
in order to assess the potential risks of OCS geoengineering for various ecosystems.

More importantly, in our understanding the negative indirect forcing due to the stratospheric aerosol produced from OCS and
its direct greenhouse gas forcing will still partially cancel each other out even at the higher concentrations. We therefore expect
the net cooling from the additional OCS to be insignificant. With that, we conclude that stratospheric aerosol enhancement by
OCS release is not a feasible climate engineering option and should not be given further thoughts.

### Code Availability

The SiB4 code is available online at https://gitlab.com/kdhaynes/sib4v2_corral. Model code on the marine OCS model is
available online at https://github.com/Sinikka-L/OCS_CS2_boxmodel. The Modular Earth Submodel System (MESSy) is con-
tinuously developed and used by a consortium of institutions. The use of MESSy and access to the source code is licensed to
all affiliates of institutions which are members of the MESSy consortium. Institutions can become a member of the MESSy
consortium by signing the MESSy Memorandum of Understanding. More information can be found on the MESSy consortium
website (https://messy-interface.org, last access: 21 December 2022). The code used for this study is based on MESSy version
2.52 stored at DKRZ and available from the authors on request.

**Data Availability**

Gridded and globally averaged SiB4 output plotted in Figures 1 & 2 is available from the authors upon request. Input data files
and model output of EMAC and code and output of the radiative convective model used here are stored at DKRZ, Hamburg.



## Author Contributions

MW and AK carried out SiB4 simulations and wrote Section 2. STL carried out ocean atmosphere exchange simulations and wrote Section 3. CB carried out radiative forcing calculations and wrote Section 4. MvH initiated the commentary and composed the manuscript. All authors discussed and edited the final manuscript.

## Competing Interests

Marc von Hobe is member of the editorial board of Atmospheric Chemistry and Physics. The peer-review process is being guided by an independent editor, and the authors have also no other competing interests to declare.

## Acknowledgements

Aleya Kaushik was supported by NOAA cooperative agreement NA22OAR4320151. Mary Whelan was supported by NASA ECOSTRESS award number 80NSSC20K0215. Sinikka Lennartz acknowledges funding from Ministry for Education and Research (BMBF, grant 16TTP079) and the Ministry for Research and Culture in Lower-Saxony (MWK 'Niedersächsisches Vorab', grant VWZN3421).

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
