# Peer review of "Comment on "An approach to sulfate geoengineering with surface emissions of carbonyl sulfide" by Quaglia et al. (2022)"

_EGUsphere, 2023_

## Author Response (AR1)

**Response to Daniele Visioni**

We thank Daniele for his openness towards the scientific discourse expressed in his opening statement. Exchanging scientific evidence and arguments, and taking them into consideration and allow conclusions to evolve, is indeed what good Science should be about. We also appreciate the important questions and comments on Section 4 of our paper, which we respond to below (in blue).

- point well taken on the IR differences between our estimates and the authors' (but I wish the authors explained more why they see such a discrepancies with our estimates in the LW. What in the radiative-convective model they use changes the results compared to ours?)

We believe that the discrepancy arises from the method chosen by Quaglia et al. to calculate the infrared forcing. For a steady state change of the mixing ratio profile of a greenhouse trace gas, the change in infrared forcing should be calculated directly and not via the GWP concept, which is designed for comparing the effect of emissions of two greenhouse gases. We have added a corresponding statement in the manuscript. We also added an additional reference that gives more detailed information on the radiative-convective model used.

- in ULAQ-CCM, evaporation is indeed calculated at every vertical level as a function of $H_2SO_4$ vapor pressure, surface area density and kernel for condensation.

We replaced "neglected" by "underestimated".

- while the authors correctly point out to Fig. S2, looking at Fig. S3 clearly highlights that the peak in extinction and SAD appears much lower than 35 km, at around 21-22km, so - while there might be discrepancies between the forcing we calculate and the authors to this piece calculate - the reason why ours is higher and theirs is lower is not the fact that the ULAQ-CCM aerosols are so higher up, or at least that's not the main factor.

Still, the sulfate maximum at ~35 km is unrealistic and also not supported by observations (as noted in the revised manuscript). Overall, the different results of the two models are consistent with findings in the model intercomparison paper by Quaglia et al. (2023), where ULAQ-CCM is near the high end and EMAC more at the low side concerning stratospheric optical depth. We added a corresponding statement and a reference to the Quaglia et al. (2023) paper in the revised manuscript.

- differences in forcing could be due to many factors, such as aerosol size (which is report in Table 1 for ULAQ-CCM), $H_2O$ changes, ozone etc.. If the overall value the authors obtain is -0.4 W/m2 that is absolutely fair, but in the conclusions to the section I would just say "which is less than a third than was estimated in Quaglia et al. (2022), and would not be able to fully compensate for the forcing by anthropogenic $CO_2$ at the suggested injection rates" rather than just "which cannot compensate for forcing by anthropogenic $CO_2$"

We changed the sentence at the end of Section 4 to: "In our calculations we get only -0.4 W m$^{-2}$ for the TTL scenario and
405     present-day conditions (2017 to 2021), which can only partially compensate for forcing by anthropogenic $CO_2$ at the suggested
injection rates."

In the conclusions, we changed "insignificant" to "insufficient" in the concluding statement on the expected net cooling from
OCS addition.

- My last comment is that it's my understanding that ACP does not accept "available from the authors upon request" as a data
410     availability statement, and so that part should be amended to include a DOI where data could be publicly accessed.

We have uploaded our data in zenodo (one record for each model) and include the appropriate references and DOIs in the
revised manuscript.

415

**Response to anonymous Referee #2**

We thank the reviewer for the appreciation of our initiative to write this short comment.
420     We are also grateful for the closing remark at the end of the review comment. We fully agree and have nothing to add, really.